# Diclofenac Diminished the Unfolded Protein Response (UPR) Induced by Tunicamycin in Human Endothelial Cells

**DOI:** 10.3390/molecules27113449

**Published:** 2022-05-26

**Authors:** Paulina Sokołowska, Małgorzata Siatkowska, Marta Jóźwiak-Bębenista, Piotr Komorowski, Marta Koptas, Edward Kowalczyk, Anna Wiktorowska-Owczarek

**Affiliations:** 1Department of Pharmacology and Toxicology, Medical University of Lodz, 7/9 Zeligowskiego, 90-752 Lodz, Poland; paulina.sokolowska@umed.lodz.pl (P.S.); marta.jozwiak-bebenista@umed.lodz.pl (M.J.-B.); edward.kowalczyk@umed.lodz.pl (E.K.); 2Laboratory of Molecular and Nanostructural Biophysics, Bionanopark, 114/116 Dubois, 93-465 Lodz, Poland; m.siatkowska@bionanopark.pl (M.S.); p.komorowski@bionanopark.pl (P.K.); 3Division of Biophysics, Institute of Materials Science and Engineering, Faculty of Mechanical Engineering, Lodz University of Technology, 1/15 Stefanowskiego, 90-924 Lodz, Poland; 4Department of Ophtalmology, Jonscher Municipal Medical Center, 14 Milionowa, 93-113 Lodz, Poland; marta.koptas@gmail.com

**Keywords:** ER stress, unfolded protein response, endothelial cells, nonsteroidal anti-inflammatory drugs (NSAIDs), CHOP/DITT3, GRP78/HSPA5

## Abstract

Diclofenac belongs to the class of nonsteroidal anti-inflammatory drugs (NSAIDs), which are amongst the most frequently prescribed drugs to treat fever, pain and inflammation. Despite the presence of NSAIDs on the pharmaceutical market for several decades, epidemiological studies have shown new clinical applications of NSAIDs, and new mechanisms of their action were discovered. The unfolded protein response (UPR) activated under endoplasmic reticulum (ER) stress is involved in the pathophysiology of many diseases and may become a drug target, therefore, the study evaluated the effects of diclofenac on the tunicamycin-induced UPR pathways in endothelial cells. RT PCR analysis showed that diclofenac significantly inhibited activation of ER stress-responsive genes, i.e., CHOP/DITT3, GRP78/HSPA5 and DNAJB9. Additionally, the drug diminished the significant upregulation and release of the GRP78 protein, as evaluated using the ELISA assay, which was likely to be involved in the mechanism of the UPR activation resulting in apoptosis induction in endothelial cells. These results suggest the value of diclofenac as a factor capable of restoring the ER homeostasis in endothelial cells by diminishing the UPR.

## 1. Introduction

The endoplasmic reticulum (ER) is an organelle that plays a major role in protein synthesis, folding, maturation, quality control and degradation [1]. When the balance between the need for protein folding and the capacity of the ER for protein folding is disturbed, due to the accumulation of misfolded and/or unfolded proteins, it leads to ER stress [2,3]. Restoring the ER homeostasis involves an activation of the unfolded protein response (UPR). The transmembrane proteins in the ER act as sensors of ER stress and they determine three major signaling pathways characterizing the UPR, i.e., the protein kinase RNA-like endoplasmic reticulum kinase (PERK), inositol-requiring enzyme 1 (IRE1) and activating transcription factor 6 (ATF-6) [1,2,3,4]. These proteins in the inactive state are bound by the ER chaperone, glucose-regulated protein 78 (GRP78), but accumulation of misfolded proteins in the ER releases the three sensors, which pass the signal to the effector proteins [1]. Depending on the conditions eliciting ER stress, the role of the UPR signaling is to promote cellular life or death [1,4]. Initially activated sensory proteins stimulate the production of pro-survival molecules and provide repair measures i.e., increased ER protein-folding capacity, degradation of misfolded proteins or reduced translation. If the protective response fails to reduce ER stress, it turns into a deleterious process due to reinforced proapoptotic signaling, leading to cell death [1,4]. The pathogenesis of many diseases, i.e., diabetes mellitus, cancer, neurodegenerative and neuropsychiatric disorders, lung disease, viral infections and others was proved to be dependent on the ER dysfunction, but the precise role of the UPR in human diseases remained an open question [4,5]. The latest data demonstrated that the GRP78 protein is translocated from the ER to the cell membrane in response to stress, where it works as a multifunctional receptor for many ligands and peptides [6,7]. Depending on the ligands or peptides that bind to the membrane GRP78, different signaling pathways can be activated leading to the induction of cell apoptosis or proliferation [6,7]. Furthermore, it was shown that overexpression of GRP78 on the cell surface rendered the stressed cell more susceptible to pathogens (viral or fungal), but the upregulation of GRP78 on the surface of cancerous cells was associated with resistance to chemotherapy [6,7]. Thus, an understanding of the molecular mechanisms of ER stress may help in the designing of new drugs or may improve the usage of already approved drugs. 

Endothelial cells lining blood vessels constitute the first barrier between the blood and the tissues in the human body that are responsible for maintaining the body’s homeostasis. Among others, the endothelial cells regulate the tension of the blood vessel walls and inhibit coagulation and platelet activation, participate in the immune system’s reactions and the formation of new blood vessels [8,9]. Endothelial dysfunction leads to the loss of vascular homeostasis and activation of the mechanisms responsible for atherosclerotic lesions, which are the cause of coronary artery disease, stroke, peripheral artery disease, renal failure and others [8,9]. The role of the UPR pathways’ activation induced by ER stress in endothelial dysfunction was highlighted in many reports [10,11,12,13], therefore the aim of the present study was to evaluate the possible role of diclofenac on the tunicamycin-induced UPR pathways in endothelial cells. Tunicamycin is a well-known ER stress inducer because it blocks the formation of protein *N*-glycosidic linkages by inhibiting the transfer of *N*-acetylglucosamine 1-phosphate to dolichol monophosphate [14].

Diclofenac belongs to the class of most frequently prescribed analgesic and anti-inflammatory drugs (NSAIDs) worldwide. Limited literature data indicate the involvement of NSAIDs in ER stress in different cellular systems [15], but it remains unknown how NSAIDs may affect ER stress in the endothelial cells. The main questions which we intended to answer in the present work were: (1) whether diclofenac can affect the tunicamycin-induced ER stress in endothelial cells and (2) how the drug can modulate the UPR signaling pathways activated under ER stress. Therefore, we investigated the effect of diclofenac on gene expression in endothelial cells following ER stress related to pro-survival, as well as the proapoptotic pathways of the UPR (Table 1).

## 2. Results

### 2.1. Effect of Diclofenac on the Viability of Endothelial Cells Undergoing ER Stress

Tunicamycin reduced the viability of endothelial cells in a dose-dependent manner (Figure 1A). In contrast, diclofenac used in a wide range of concentrations, i.e., 1–100 µM, did not induce any changes in cell viability (Figure 1B). For further experiments, tunicamycin at a concentration of 0.5 µg/mL was chosen because the drug evoked a significant but not too high decrease in cell viability, in a range of 70–85%, to avoid the cytotoxic effects of the ER stress inducer.

Concomitant 24-h treatment of endothelial cells with diclofenac (1–100 µM) and tunicamycin (0.5 µg/mL) caused a dose-dependent increase in the cell viability (Figure 2). Diclofenac at concentrations ranging from 25 to 75 µM evoked a statistically significant rise in cell viability of the tunicamycin-treated cells, with the highest mean value of 97.5% observed at 75 µM. In comparison, the viability of tunicamycin-treated cells was decreased to 84.5%.

### 2.2. Effect of Diclofenac on Gene Expression Related to ER Stress in Endothelial Cells

After 24-h incubation of the endothelial cells with tunicamycin (0.5 µg/mL), five genes showed an altered expression at the 1.5 fold change cut-off. Among them, CHOP/DITT3, GRP78/HSPA5 and DNAJB9 were significantly upregulated, with fold changes of 21.9, 19.0, and 11.6, respectively, while ATF6 and ATF4 gene expression exceed the 1.5-fold change value (Figure 3). Tunicamycin did not change the expression of other tested genes. It was found that 24-h incubation of the tunicamycin-treated endothelial cells with diclofenac (75 µM), significantly reduced expression of CHOP/DITT3 (fold change of 21.9 reduced to 3.1), GRP78/HSPA5 (fold change of 19.1 reduced to 5.1) and DNAJB9 (fold change of 11.6 reduced to 1.6) compared to the cells treated with tunicamycin alone (Figure 3).

### 2.3. Effect of Diclofenac on GRP78 and CHOP Proteins Expressions and Apoptosis Induction in Endothelial Cells Undergoing ER Stress

The 24-h incubation of the endothelial cells with tunicamycin (0.5 µg/mL) evoked a significant rise in the expression and release of GRP78 protein by the endothelial cells, whereas the simultaneous addition of diclofenac (75 µM) to tunicamycin-treated cells in a statistically significant manner diminished the protein level in the cell supernatants (693% vs. 363% of control for tunicamycin and diclofenac with tunicamycin, respectively) (Figure 4A). In contrast, diclofenac did not alter significantly the expression of the CHOP protein; only a slight tendency to diminish the CHOP protein expression can be observed (270% vs. 203% of control for tunicamycin and diclofenac with tunicamycin, respectively) (Figure 4B). We also checked how diclofenac may modulate the apoptosis of the endothelial cells. As measured by annexin V labelling, after 24 h, a low but statistically significant rise of the apoptotic cells was observed in tunicamycin-treated cells (Figure 5). Concomitantly added diclofenac decreased the percentage of apoptotic cells in a statistically significant manner, and to the similar level as detected in the untreated control cells (Figure 5).

## 3. Discussion

The best recognized mechanism of the action of NSAIDs is the inhibition of two different isoforms of cyclooxygenase (COX) enzymes. Inhibition of COX-2 is related to the anti-inflammatory, analgesic and antipyretic actions of NSAIDs, whereas the blockade of COX-1 is partly responsible for gastric toxicity [15]. Prolonged use of NSAIDs may evoke severe lesions in the gastrointestinal (GI) tract such as ulceration, bleeding or perforations [15]. In a search for the mechanisms underlying the cytotoxicity of NSAIDs in the gastric mucosa, the ER stress response was shown to be an important factor involved in apoptosis induction and the GI injuries evoked by these drugs [15]. Treatment (16 h) with diclofenac (1 mM), and other NSAIDs decreased the viability of cultured gastric mucosal cells of guinea pigs, along with increased DNA fragmentation and elevated amounts of CHOP mRNA [18]. In those studies, diclofenac elicited ER stress-dependent apoptosis of cultured gastric mucosal cells [18]. In contrast, our report showed an opposite, antiapoptotic effect of diclofenac on human endothelial cells. One possible explanation of that difference is the higher concentration of diclofenac used in the studies on gastric mucosal cells, i.e., 1 mM, at which diclofenac evoked a significant cell cytotoxicity (cell viability was less than 50%) [18]. In our studies, we tested the effect of diclofenac in the concentration range of 0.001–0.1 mM for 24 h on the viability of endothelial EA.hy 926 cells and the drug did not elicit any cytotoxicity, even at the highest concentration tested, i.e., 0.1 mM (Figure 1B). However, other studies have shown that a much lower concentration than 1 mM, i.e., 0.1 mM of diclofenac administered for 6 h, increased the apoptotic DNA fragmentation in cultured gastric mucosal cells [19], and 0.45 mM concentration of this drug led to hepatotoxicity [20]. Thus, another explanation could be the much higher sensitivity of gastric mucosal cells or hepatocytes to diclofenac and other NSAIDs than that of the endothelial cells.

Diclofenac induced apoptosis not only in the primary gastric mucosal cells but also in rat hepatocytes [20,21], and Huh7 hepatoma cell line [22]. Hepatotoxicity is also one of the adverse effects associated with the administration of diclofenac. In rat hepatocytes, the UPR signaling molecule involved in the proapoptotic actions of diclofenac was the CHOP protein [21], but in the Huh7 hepatoma cells the drug also reduced the expression of ATF6 and evoked phosphorylation of eIF2alpha, a protein directly activated by PERK [22]. Prolonged activation of the PERK-eIF2alpha pathway could induce expression of the CHOP protein, which was also shown to be involved in diclofenac-induced apoptosis in Huh7 hepatoma cells. The authors of the work concluded that diclofenac elicited upregulation of the proapoptotic PERK-eIF2alpha-CHOP pathway and simultaneously reduced the expression of the pro-survival ATF6 sensor protein, and did not stimulate IRE1 endonuclease, which drives the expression of pro-survival factor XBP1 [22]. Thus, upregulation of the proapoptotic UPR pathway by diclofenac was related to a diminished or unaltered expression of the pro-survival UPR pathways.

On the other hand, Yamazaki and colleagues [23] demonstrated an opposite response of diclofenac and other NSAIDs in human neuroblastoma SHSY5Y cells challenged with the ER stress inducers, tunicamycin or thapsigargin. Diclofenac, at concentrations of 1–300 µM, added 2 h before the 24-h stimulation by thapsigargin or tunicamycin, significantly and in a concentration-dependent manner increased the cell viability. The pro-survival action of the drug resulted from the apoptosis suppression of neuroblastoma SHSY5Y cells evoked by the ER stress inducers. The mechanism underlying the antiapoptotic effect of diclofenac was related to the inhibition of caspase-2, -9 and -3 activation and prevention of a decrease in mitochondrial membrane potential caused by ER stress [23]. Results of our studies corresponded to those of Yamazaki and colleagues, since we demonstrated a pro-survival effect of diclofenac on endothelial cells. We observed the action of diclofenac in a lower range of concentrations, i.e., 25–75 µM, with the highest effect at 75 µM. It may be associated with differences in the type of cells examined. Interestingly, despite induction of apoptosis by tunicamycin and significant overexpression of the *CHOP* gene, we did not observe any changes in the expression of genes encoding the BCL-2 protein family responsible for activation of the caspase cascade. It is worth mentioning that *GADD153/CHOP* is a transcription factor which downregulates the expression of Bcl-2 and upregulates the expression of other proapoptotic members of the Bcl-2 family [24,25]. The phenomenon observed in our studies may result from the simultaneous high expression of chaperone GRP78, which was shown to interfere with caspase activation [26,27].

In contrast to the studies of Yamazaki and colleagues [23], who used the ER stress inducers to elicit cell cytotoxicity, the goal of our work was to determine the concentration of tunicamycin which evoked reduction in cell viability to the values less than cytotoxic, to avoid a too intense UPR activation with only apoptotic and/or necrotic events. This methodology enabled us to evaluate the contribution of diclofenac to the expression of UPR genes activated by tunicamycin in endothelial cells. Diclofenac significantly decreased the tunicamycin-induced overexpression of *CHOP/DITT3*, *GRP78/HSPA5*, and *DNAJB9* genes (Figure 3). In the next stage of the study, the effects of diclofenac on GRP78 protein release and CHOP protein expression in tunicamycin-treated endothelial cells were evaluated. It was demonstrated that diclofenac significantly diminished the GRP78 protein level but surprisingly had only a minor and statistically insignificant influence on upregulation of the CHOP protein, which is a major regulator of apoptosis during ER stress (Figure 4). However, diclofenac reduced a significant (four-fold) increase in the percentage of apoptotic cells evoked by tunicamycin (Figure 5). The discrepancy between the effect of diclofenac on tunicamycin-induced CHOP expression at gene and protein level may result from the timing of induction of the different UPR pathways and their mutual interactions, which determine the final cell response to ER stress. In the light of these findings it also seems possible that such a significant overexpression of GRP78 at the gene and protein level might be implicated in apoptosis induction. Under tunicamycin-induced ER stress in endothelial cells, GRP78 protein might be translocated to the cell membrane. According to the latest data, when GRP78 is located on the cell surface it acts as a receptor for many ligands, therefore different signaling cascades can be activated resulting in the occurrence of opposite events, such as the induction of cell apoptosis or proliferation [6,7]. Results of the study suggest that tunicamycin evoked a significant overexpression of the GRP78 gene followed by a massive release of the protein, resulting, at least partly, in apoptosis induction in the endothelial cells. Concomitant addition of diclofenac significantly reduced the percentage of apoptotic cells induced by tunicamycin to nearly control values. Diclofenac’s mechanism of action involves the inhibition of the *CHOP/DITT3*, *GRP78/HSPA5* and *DNAJB9* genes’ overexpression and the reduction of the GRP78 protein release, resulting in increased viability of endothelial cells. However, we would like to emphasize that we have shown only a mechanistic study, because the used concentrations of diclofenac are not observed in the vascular system in patients. Nevertheless, our work confirms that the effects of diclofenac and probably other NSAIDs may result not only from their anti-inflammatory properties, but also that other signaling pathways seem to be involved. therefore further investigations are required in this regard.

## 4. Materials and Methods

### 4.1. Reagents

Tunicamycin, diclofenac, MTT (3-(4,5-dimethylthiazol-2-yl)-2,5-diphenyltetrazolium bromide) and Hoechst 33342 solution were purchased from Sigma-Aldrich (Saint Louis, MO, USA). Dulbecco’s Modified Eagle Medium (DMEM), Fetal Bovine Serum (FBS), Dulbecco’s Phosphate Buffered Saline (D-PBS) and Trypsin-EDTA solution were obtained from Biowest (Nuaillé, France). The Annexin V-FITC Fluorescence Microscopy Kit was obtained from BD Pharmingen (Franklin Lakes, NJ, USA), the Human HSPA5/ Endoplasmic reticulum chaperone BiP ELISA Kit was obtained from EIAab Science (Wuhan, China), the RNeasy Mini Kit was obtained from Qiagen (Germantown, WI, USA) and the High Sensitivity RNA ScreenTape Kit was obtained from Agilent Technologies (Santa Clara, CA, USA). Reagents for Real-Time PCR, i.e., Custom PrimePCR™ Real-Time PCR Plates, iScript™ cDNA Synthesis Kit, 2xSsoAdvanced Universal SYBR Green Supermix, Prime PCR RT Control and Prime PCR Control Assay, as well as the reagents for Western blotting, i.e., Mini-Protean TGX Stain-Free Gels, 10x Tris/Glycine/SDS Buffer, Trans-Blot Turbo RTA Transfer Kit, Nitrocellulose, Trans-Blot Turbo 5x Transfer buffer, Clarity Western ECL Substrate Assay, 4x Laemmli Sample Buffer, 2-Mercaptoethanol, Precision Plus Protein All Blue Standards and Precision Plus Protein Unstained Standards were all purchased from Bio-Rad (Berkeley, CA, USA). Reagents for the lysis buffer (urea, thiourea, Tris, 3-[(3-Cholamidopropyl)dimethylammonio]-1-propanesulfonate hydrate (CHAPS), IPG Buffer pH 4–7, dithiotreitol (DTT)), 2D Quant Kit and 2D Clean Up Kit were purchased from GE Healthcare (Little Chalfont, UK). Anti-DDIT3 and anti-mouse IgG (HRP) were purchased from Abcam (Cambridge, MA, USA).

### 4.2. Cell Culture

The studies were performed on a commercially available EA.hy926 endothelial cell line (ATCC CRL-2922). Cell culture was maintained at 37 °C and in a humidified atmosphere of 5% CO_2_. Cells were grown in DMEM supplemented with 10% of FBS, according to the protocol recommended by American Type Culture Collection (ATCC).

### 4.3. MTT Test

The viability of the EA.hy926 cells was tested using the colorimetric MTT assay. The cells were seeded onto a 96-well plate to a final density of 8 × 10^3^ cells/well. After 24 h of culture the cells were exposed to tunicamycin or tunicamycin with diclofenac for the next 24 h. After treatment with drugs, MTT solution was added to the cell culture for another 4 h. The value of absorbance, proportional to the number of viable cells, was measured at 570 nm by microplate spectrophotometer (Victor X4, Perkin Elmer, Waltham, MA, USA). The viability was calculated as follows: Viability (%) = (A/AC) × 100%; where A is the absorbance of an investigated sample; AC is the absorbance of control (untreated). For the selection of proper concentrations for tunicamycin and diclofenac used in all of the experiments, preliminary tests assessing cell viability were performed for concentrations ranging from 0.1–40 µg/mL and 1–100 µM for tunicamycin and diclofenac, respectively.

### 4.4. Gene Expression Analysis

Endothelial EA.hy926 cells were seeded onto culture flasks at a density of 1 × 10^6^ cells/flask and cultured for 24 h. Next, tunicamycin (0.5 µg/mL) or tunicamycin concomitantly with diclofenac (75 µM) were added to the culture for 24 h. After the incubation with drugs, cells were washed with D-PBS, trypsinized and centrifuged at 150× *g*, 5 min, room temperature (RT). The cell pellets were resuspended in RLT lysis buffer and subjected to RNA isolation and column purification using the RNeasy Mini Kit from Qiagen. The RNA concentration in the samples was measured with the NanoVue Plus spectrophotometer (GE Healthcare), and the RNA integrity was determined by electrophoresis using the High Sensitivity RNA ScreenTape Kit from Agilent Technologies (Santa Clara, CA, USA). All of the samples taken for further processing were characterized by RNA Integrity Number (RIN) above nine. A reverse transcription reaction was performed for 1 µg RNA in a SureCycler 8800 thermocycler (Agilent Santa Clara, CA, USA) using the iScript™ cDNA Synthesis Kit containing reverse transcriptase and a blend of oligo(dT) and random hexamer primers. The cDNA of each sample with a concentration of 10 ng/µL, the 2xSsoAdvanced Universal SYBR Green Supermix reagent, containing, i.e., dNTPs mixture, thermostable polymerase and SYBR Green marker, were added into appropriate wells on the 96-well defined Custom PrimePCR™ Real-Time PCR Plate, according to the manufacturer’s protocol (Bio-Rad). In addition, the PCR reaction control (Prime PCR Control Assay, Bio-Rad, Berkeley, CA, USA) and RT reaction control (Prime PCR RT Control) were included in all of the experiments. The real-time PCR reaction was carried out in a CFX96 thermal cycler (Bio-Rad, Berkeley, CA, USA) and the results were analyzed with the 2^−ΔΔCq^ method. The reference genes, i.e., *GAPDH* and *TBP*, were used for normalization of the results. Statistical significance was assessed with one-way ANOVA. Fold change value above 1.5 was chosen as the cut-off criterion, while statistical significance was assumed at *p* < 0.05.

### 4.5. GRP78 Protein Release

The release of the GRP78 protein was measured with the Human HSPA5/ Endoplasmic reticulum chaperone BiP ELISA Kit. The EA.hy926 cells were seeded onto a 12-well plate to a final density of 1 × 10^5^ cells/well. After 24 h of culture, the cells were exposed to tunicamycin (0.5 µg/mL) or tunicamycin with diclofenac (75 µM) for the next 24 h. After 24 h of treatment with drugs, the cell supernatants were collected, centrifuged at 250× *g* for 5 min to avoid cellular debris, subjected to analysis according to the manufacturer’s protocol, and measured using the microplate reader BioTek ELx800 (BioTek, Winooski, VT, USA).

### 4.6. CHOP Protein Expression Analysis

The CHOP protein expression was measured using immunoblotting. For the experiments, cells were seeded onto culture flasks to a final density of 1 × 10^6^ cells. After 24 h of culture, the cells were exposed to tunicamycin (0.5 µg/mL) or tunicamycin with diclofenac (75 µM) for another 24 h. After the treatments, the cells were washed in PBS, trypsinized, and centrifuged at 150× *g* for 5 min at RT. Cell pellets were resuspended in lysis buffer (7 M urea, 2 M thiourea, 4% *w*/*v* CHAPS, 2% *v*/*v* IPG buffer, 40 mM DTT). Concentrations of proteins were measured by 2D Quant kit, according to the manufacturer’s instructions. To remove all of the interfering compounds, a clean-up procedure was performed with the 2D Clean Up Kit, according to the manufacturer’s protocol. Electrophoresis and Western blotting were performed using stain-free technology (Bio-Rad, Berkeley, CA, USA). Total protein normalization is usually based on the use of housekeeping proteins, but in the experiments the stain-free total protein normalization was used, according to the technology available in Bio-Rad, which enables quantification of the abundance of the protein of interest without relying on housekeeping proteins. The technology uses a trihalo compound that is directly incorporated into the gel (pre-cast stain-free gels were purchased from Bio-rad, Berkeley, CA, USA). Upon UV exposure, the compound modifies the tryptophan residues in proteins, causing them to fluoresce. Proteins can be visualized in the gel before and after the transfer onto the membrane. Therefore, sample integrity and separation quality after electrophoresis was verified with Gel_Doc XR+ Imager. The proteins were then transferred to nitrocellulose membranes using the Trans-blot Turbo system (Bio-Rad, Berkeley, CA, USA). Transfer efficiency was determined using a Gel_Doc XR+ Imager on the post-transfer gels and blots. Thereafter, the membranes were blocked in 3% (*w*/*v*) non-fat dry milk in TBST (Tris-buffered saline containing 0.1% Tween), for 1 h at RT. After overnight incubation at 4 °C with the primary antibody (Anti-DDIT3, 1:250, ab 11419, Abcam, state, city, country Cambridge, MA, USA), membranes were washed several times in TBST, incubated with secondary antibodies at room temperature for 1 h (anti-mouse IgG (HRP), 1:4000, ab205719, Abcam, Cambridge, MA, USA), and washed as described before. Chemiluminescence was performed using the Clarity Max Western ECL Substrate kit and detected by ChemiDoc MP imaging system (Bio-Rad, Berkeley, CA, USA). Data were analyzed by Image Lab (version 5.2.1, Bio-Rad, Berkeley, CA, USA). The results were normalized to total protein level according to the manufacturer’s protocol (Stain-free Western-blotting protocol, Bio-Rad). Briefly, stain-free total protein normalization was performed by measuring the total protein signal directly on the membrane, which was visualized before the chemiluminescent reaction and after the staining. In the Image Lab software, we downloaded both of the images and created a multichannel image of the stain-free and chemiluminescent blot images. Image Lab software automatically detected lanes and bands. We selected a lane used for normalization (control sample, without treatment) and compared the chemiluminescent signal in each lane to its corresponding stain-free lane, thus, the normalized volume value represents each chemiluminescent signal that is properly the total protein normalized to the stain-free signal in the corresponding lane.

### 4.7. Annexin V Labelling

Apoptosis in EA.hy926 cells was detected with the Annexin V-FITC Fluorescence Microscopy Kit. The assay is based on the interaction of Annexin V, labelled with FITC fluorochrome, with phosphatidylserine (PS), exposed during the early stages of apoptosis. For the experiments, cells were seeded onto a 96-well plate to a final density of 8 × 10^3^ cells/well. After 24 h of culture, cells were exposed to tunicamycin (0.5 µg/mL) or tunicamycin with diclofenac (75 µM) for another 24 h. After incubation, cells were rinsed with D-PBS and stained with Annexin V-FITC, according to the manufacturer’s protocol. Additionally, 1 µg/mL Hoechst 33342 solution was used to stain the cell nuclei. Labelled cells were visualized with the use of the InCell Analyzer 2000 (GE Healthcare) and analyzed by dedicated InCell Development software. Apoptotic cells were identified on the basis of the fluorescence intensity observed for the FITC channel, while the total numbers of cells, both live and apoptotic, were estimated according to Hoechst staining.

### 4.8. Data Analysis

Data are expressed as mean ± standard deviation (SD). The number of experiments for each method (*n*) is given below the figure. The results were tested by one-way ANOVA followed by post hoc Tukey’s multiple comparisons test. All of the calculations were performed using GraphPad InStat version 9.3.0 (GraphPad, San Diego, CA, USA).

## 5. Conclusions

Tunicamycin-induced ER stress in endothelial cells resulted in the activation of the UPR signaling with the simultaneous initiation of both proapoptotic and pro-survival genes, but the final outcome of interactions between the UPR pathways resulted in the significant upregulation and release of the GRP78 chaperone and apoptosis induction, with a concomitant decrease in viability of endothelial cells.

Diclofenac increased the endothelial cells’ survival by inhibiting the UPR signaling pathways activated by tunicamycin.

These results suggest the possible role of diclofenac as a factor capable of restoring the ER homeostasis in endothelial cells by diminishing the unfolded protein response, but the precise mechanism of action of diclofenac needs to be investigated.

## Figures and Tables

**Figure 1 molecules-27-03449-f001:**
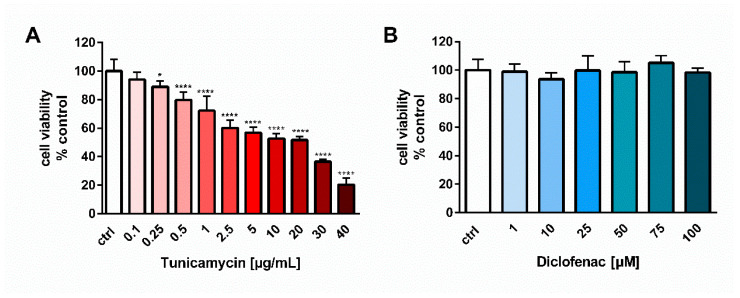
Effect of tunicamycin (**A**) and diclofenac (**B**) on the viability of endothelial cells evaluated using MTT assay. Data are presented as mean ± SD and expressed as percentage of untreated control cells (*n* = 4). Statistical significance vs. control cells is indicated when appropriate; * *p* < 0.05; **** *p* < 0.0001. The results were tested by one-way ANOVA followed by post hoc Tukey’s multiple comparisons test.

**Figure 2 molecules-27-03449-f002:**
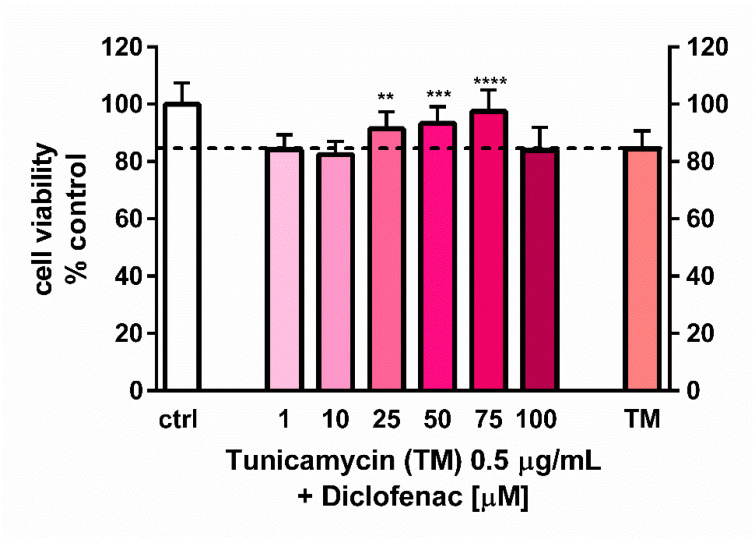
Effect of diclofenac on the viability of tunicamycin-treated endothelial cells evaluated using MTT assay. Data are presented as mean ± SD and expressed as percentage of untreated control cells (*n* = 4) Statistical significance vs. control cells is indicated when appropriate; ** *p* < 0.05; *** *p* < 0.001; **** *p* < 0.0001. The results were tested by one-way ANOVA followed by post hoc Tukey’s multiple comparisons test.

**Figure 3 molecules-27-03449-f003:**
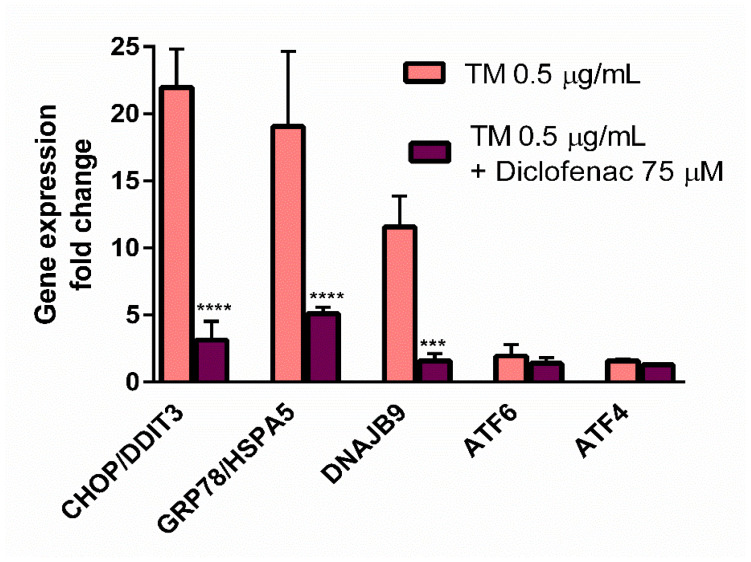
Effect of diclofenac on gene expression of tunicamycin-treated endothelial cells evaluated using RT-PCR. Data are presented as mean ± SD and expressed as fold change vs. untreated control cells (*n* = 3). Statistical significance vs. tunicamycin-treated cells is indicated when appropriate; *** *p* < 0.001; **** *p* < 0.0001. The results were tested by one-way ANOVA followed by post hoc Tukey’s multiple comparisons test.

**Figure 4 molecules-27-03449-f004:**
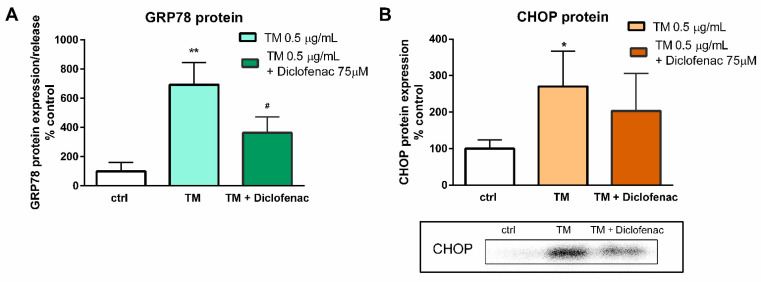
Effect of diclofenac on expression/release of GRP78 evaluated using ELISA assay (**A**) and expression of CHOP evaluated using Western blot (**B**) proteins of tunicamycin-treated endothelial cells. Total protein normalization was performed without housekeeping proteins using stain-free total protein normalization technology. Data are presented as mean ± SD and expressed as percentage of untreated control cells (A *n* = 3; B *n* = 4). Statistical significance vs. control cells is indicated when appropriate; * *p* < 0.05 vs. untreated control ** *p* < 0.01 vs. untreated control; # *p* < 0.05 vs. tunicamycin-treated cells. The results were tested by one-way ANOVA followed by post hoc Tukey’s multiple comparisons test.

**Figure 5 molecules-27-03449-f005:**
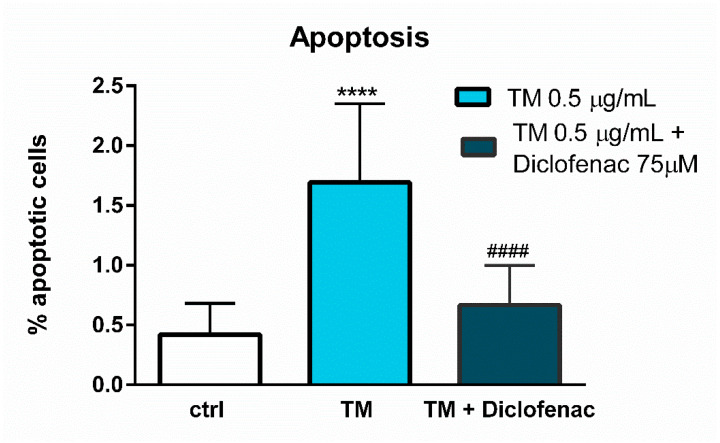
Effect of diclofenac on apoptosis evoked by tunicamycin in endothelial cells evaluated using the Annexin V test. Data are presented as mean ± SD and expressed as a percentage of untreated control cells (*n* = 3). Statistical significance vs. control cells is indicated when appropriate; **** *p* < 0.0001 vs. untreated control; #### *p* < 0.0001 vs. tunicamycin-treated cell. The results were tested by one-way ANOVA followed by post hoc Tukey’s multiple comparisons test.

**Table 1 molecules-27-03449-t001:** Genes and their function in ER stress signaling chosen for Real-Time PCR experiments.

Gene Symbol	Gene Name	Function in ER Stress Signaling	Reference
*ATF4*	Activating transcription factor 4	Downstream target of PERK, which drives expression of adaptive, pro-survival genes. Under sustained stress it also activates genes involved in cell death	[1]
*ATF6*	Activating transcription factor 6	A sensor of ER stress, activates transcription of genes maintaining ER homeostasis, i.e., gene expression of ER chaperones, ERAD components	[5]
*ATF6B*	Activating transcription factor 6 beta	An inhibitor of UPR genes induced by ATF6α	[1,16]
*BAX*	BCL2 associated X gene	Controls the release of cytochrome c during activation of ER-related mitochondrial apoptosis	[1]
*BCL2*	B-cell lymphoma 2	Regulates store-operated Ca ^2+^ entry to prevent ER stress-induced apoptosis	[1]
*CHOP/* *DITT3*	C/EBP homologous protein/ DNA damage inducible transcript 3	Main transcription factor involved in regulation of apoptosis during ER stress. The ATF4-CHOP pathway increases the expression of other proapoptotic genes	[1]
*DNAJB9*	DnaJ heat shock protein family (Hsp40) member B9	The ER luminal co-chaperone inhibits IRE1 by promoting a complex between GRP78 and the luminal stress sensing domain of IRE1α.	[17]
*EIF2alpha*	Eukaryotic translation initiation factor 2 alpha	Directly activated by PERK (a sensor of ER stress), causes general inhibition of protein synthesis and promotes ATF4 protein production. It can also directly enhance the translation of CHOP	[1]
*ERN1*	Endoplasmic reticulum to nucleus signaling 1	Encodes inositol-requiring enzyme 1 α (IRE1α)—a sensor of ER stress. IRE1-mediated splicing of XBP1 mRNA results in transcription of pro-survival factors	[1]
*GRP78/HSPA5*	Heat shock protein family (Hsp70) member 5	The ER chaperone, plays a role in fine-tuning of the UPR—reduces ER stress and increases cell survival. When translocated to the cell surface it acts as a multifunctional receptor.	[1,5,7]
*MAPK8*	Mitogen-activated protein kinase 8/ (c-Jun N-amino-terminal kinase JNK1)	Regulates several BCL-2 family members, including the activation of proapoptotic BID and BIM, and inhibition of antiapoptotic BCL-2, BCL-XL and MCL-1	[1]
*TRAF2*	TNF receptor-associated factor 2	Activates apoptosis signal-regulating kinase 1 (ASK1) and its downstream targets c-Jun NH2-terminal kinase (JNK) and p38 MAPK	[1]

## Data Availability

Not applicable.

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
