# Peer review of "Diclofenac Diminished the Unfolded Protein Response (UPR) Induced by Tunicamycin in Human Endothelial Cells"

_molecules, 2022, doi:10.3390/molecules27113449_

Round 1
Reviewer 1 Report
This manuscript by Sokołowska et al. presented a study of the effect of diclofenac on tunicamycin-induced Unfolded Protein Response (UPR) in human endothelial cells. The authors found that diclofenac reduced the expression of CHOP/DITT3, GRP78/HSPA5, and DNAJB9 genes, as well as GRP78 protein expression, but it has a minor effect on CHOP protein. As a result, diclofenac increased the viability of human endothelial cells.
Overall this study is well designed and the results are clearly presented. It is appropriate for publication in Molecules. My only concern is that the authors should have more discussion on the difference between the current work and previous work regarding the effect of diclofenac on apoptosis.
In Line 145 through 164, the author offered two explanations: (1) a higher concentration of diclofenac, 1 mM, was used in previous work; (2) gastric mucosal cells may have higher sensitivity than other cells.
However, several studies showed that a lower concentration of diclofenac could also induce apoptosis.
Kusuhara et al. (https://doi.org/10.1016/S0014-2999(98)00679-7) showed that treatment with 0.1 mM diclofenac for 6 hours could increase the apoptotic DNA fragmentation.
Gómez-Lechón demonstrated the apoptotic effect of diclofenac under sub-cytotoxic concentration (0.45 mM) in hepatocytes.
In addition, a review article (https://doi.org/10.1185/03007995.2010.486301) summarized that under certain conditions, diclofenac may promote inflammation by "inhibiting PPARg signaling by acting as a competitive antagonist".
These points should be discussed in the revised manuscript.
There are also some typos/ grammatical errors.
Line 126-127: "only a slight tendency do diminish". "do" should be "to".
Line 193: "It is worth to mention that" should be "it is noteworthy that" or "it is worth mentioning that".
Author Response
Manuscript Number: molecules-1696198
Title: Diclofenac diminished the unfolded protein response (UPR) induced by tunicamycin in human endothelial cells.
Reviewer #1:
This manuscript by Sokołowska et al. presented a study of the effect of diclofenac on tunicamycin-induced Unfolded Protein Response (UPR) in human endothelial cells. The authors found that diclofenac reduced the expression of CHOP/DITT3, GRP78/HSPA5, and DNAJB9 genes, as well as GRP78 protein expression, but it has a minor effect on CHOP protein. As a result, diclofenac increased the viability of human endothelial cells.
Overall this study is well designed and the results are clearly presented. It is appropriate for publication in Molecules. My only concern is that the authors should have more discussion on the difference between the current work and previous work regarding the effect of diclofenac on apoptosis.
At the very beginning we would like to thank the Reviewer for the great commitment and careful revision of our manuscript. We are convinced that all suggestions improve the quality of our paper. Revisions to the manuscript are marked up using the “Track Changes”
In Line 145 through 164, the author offered two explanations: (1) a higher concentration of diclofenac, 1 mM, was used in previous work; (2) gastric mucosal cells may have higher sensitivity than other cells.
However, several studies showed that a lower concentration of diclofenac could also induce apoptosis.
Kusuhara et al. (https://doi.org/10.1016/S0014-2999(98)00679-7) showed that treatment with 0.1 mM diclofenac for 6 hours could increase the apoptotic DNA fragmentation.
Gómez-Lechón demonstrated the apoptotic effect of diclofenac under sub-cytotoxic concentration (0.45 mM) in hepatocytes.
In addition, a review article (https://doi.org/10.1185/03007995.2010.486301) summarized that under certain conditions, diclofenac may promote inflammation by "inhibiting PPARg signaling by acting as a competitive antagonist".
These points should be discussed in the revised manuscript.
We added suggested two first literature data in the revised manuscript. However, the suggested review article, which we found to be very interesting, seems to be beyond the scope of our short communication that is focused on the discussion of the effects of diclofenac on the UPR signalling pathways. In case of our future publication we will take into consideration this review to make a nice contribution into further discussion of the effect of diclofenac on other signalling pathways.
There are also some typos/ grammatical errors.
Line 126-127: "only a slight tendency do diminish". "do" should be "to".
Line 193: "It is worth to mention that" should be "it is noteworthy that" or "it is worth mentioning that".
Errors have been corrected.

Reviewer 2 Report
Review of the manuscript molecules-1696198
The authors report on the effects of diclofenac on tunicamycin-induced endoplasmatic reticulum (ER) stress. The ER stress results in a change in expression and relocation of the unfolded proteins controlled by the ER-chaperone GRP78. This is an interesting study as it shows the possible
mechanistic impact of diclofenac as an NSAID on ER stress even if due to the used concentration it is a positive effect protecting the endothelium. It is not exactly clear what the authors meant if someone reads the first sentence of the abstract as many people would only think of the negative side effects of NSAIDs and not really the positive ones. As the authors discuss, higher
concentrations of diclofenac seem to turn the drug into an ER stress inducer. Well, this brings me to my main criticism of the study and that is the concentration used in this study which is, of course, compared to other studies on the low side but way out of range in terms of the real
concentration in a patient. I will explain this in detail later. It would mean that this study could only be considered as an in vitro study with no real implications for patients. The methods are sound, and the results are presented in a clear and professional way supporting the discussion.
In general, a clear story with an interesting aspect for ER stress experts even if the translational aspect is more than questionable. I have some comments the authors should consider:
1. So, I will now explain why this study is way out of range in terms of the used diclofenac concentration. According to a recent study using a 100 mg single dose the peak plasma concentration is 625 ng/mL. This concentration based on the mw for diclofenac of 296 g/mol gives approx. 2 μM (Shah et al. Chemistry Central Journal (2016) 10:52). Even if we would calculate the maximum dose of 200 mg per day this would possibly only amount to
4 μM in plasma as a peak value. Another recent study documented the plasma concentration of diclofenac based on a 37,5 mg oral dose of diclofenac at 4.4 – 6.63 ng/mL which equates to 15 – 22 nM (Miyatake et al. Br J Clin Pharmacol. 2009 Jan; 67(1): 125–129). And again, if we consider 200 mg this would only give 75 – 100 nM. I guess the authors get my point that patient plasma concentrations will never reach the applied effective concentrations the authors have used of 75 μM diclofenac. The authors have not seen any effects of diclofenac below 25 μM. This fact will make this study completely irrelevant for patients. It could be documented as a mechanistic study if the authors clearly state in the discussion that the used concentrations of diclofenac are not seen in patients, and that the effects of diclofenac seen in this study have no implications for the situation in the vascular system in patients.
Minor points:
2. Can I ask the authors to provide the blot they used for total protein normalisation in figure 4B? It is standard procedure to show the blot.
3. All figure legends require the number of experiments done (n=?) and the statistical test
used.
Author Response
Manuscript Number: molecules-1696198
Title: Diclofenac diminished the unfolded protein response (UPR) induced by tunicamycin in human endothelial cells.
At the very beginning we would like to thank the Reviewer for the great commitment and accurate comments concerning our manuscript. Revisions to the manuscript are marked up using the “Track Changes”.
Review of the manuscript molecules-1696198
The authors report on the effects of diclofenac on tunicamycin-induced endoplasmatic reticulum (ER) stress. The ER stress results in a change in expression and relocation of the unfolded proteins controlled by the ER-chaperone GRP78. This is an interesting study as it shows the possible
mechanistic impact of diclofenac as an NSAID on ER stress even if due to the used concentration it is a positive effect protecting the endothelium. It is not exactly clear what the authors meant if someone reads the first sentence of the abstract as many people would only think of the negative side effects of NSAIDs and not really the positive ones.
We proposed other beginning of the abstract to show goals of our study more clear for the readers.
As the authors discuss, higher concentrations of diclofenac seem to turn the drug into an ER stress inducer. Well, this brings me to my main criticism of the study and that is the concentration used in this study which is, of course, compared to other studies on the low side but way out of range in terms of the real concentration in a patient. I will explain this in detail later. It would mean that this study could only be considered as an in vitro study with no real implications for patients. The methods are sound, and the results are presented in a clear and professional way supporting the discussion.
In general, a clear story with an interesting aspect for ER stress experts even if the translational aspect is more than questionable. I have some comments the authors should consider:
1. So, I will now explain why this study is way out of range in terms of the used diclofenac concentration. According to a recent study using a 100 mg single dose the peak plasma concentration is 625 ng/mL. This concentration based on the mw for diclofenac of 296 g/mol gives approx. 2 μM (Shah et al. Chemistry Central Journal (2016) 10:52). Even if we would calculate the maximum dose of 200 mg per day this would possibly only amount to
4 μM in plasma as a peak value. Another recent study documented the plasma concentration of diclofenac based on a 37,5 mg oral dose of diclofenac at 4.4 – 6.63 ng/mL which equates to 15 – 22 nM (Miyatake et al. Br J Clin Pharmacol. 2009 Jan; 67(1): 125–129). And again, if we consider 200 mg this would only give 75 – 100 nM. I guess the authors get my point that patient plasma concentrations will never reach the applied effective concentrations the authors have used of 75 μM diclofenac. The authors have not seen any effects of diclofenac below 25 μM. This fact will make this study completely irrelevant for patients. It could be documented as a mechanistic study if the authors clearly state in the discussion that the used concentrations of diclofenac are not seen in patients, and that the effects of diclofenac seen in this study have no implications for the situation in the vascular system in patients.
We agree entirely with the Reviewer that results of our work and some other in vitro studies to which we referred (e.g. 21-23] are not useful clinically. We added a sentence in the discussion emphasising this fact (lines 242-247). We also removed the sentence in conclusions which could suggest possible implications of our results for patients (lines 388-390). We think this is an important issue concerning in vitro studies in general. Therefore, we are going to design our next studies not only on the basis of the literature data but also we would like to take into consideration their possible translation into clinical application.
Minor points:
- Can I ask the authors to provide the blot they used for total protein normalisation in figure 4B? It is standard procedure to show the blot.
The blots were attached. We would like to explain that traditionally total protein normalization is based on the use of housekeeping proteins. However, in our experiments we used Stain-Free total protein normalization according to the technology available in Bio-Rad which enables quantification of the abundance of the protein of interest without relying on housekeeping proteins. The technology uses a trihalo compound that is directly incorporated into the gel (Pre-cast stain-free gels were purchased from Bio-rad). Upon UV exposure, the compound modifies tryptophan residues in proteins causing them to fluoresce. The fluorescent signal is then detected by a CCD camera. Proteins were visualized in the gel before transfer and after transfer on the membrane. Stain-Free total protein normalization was performed by measuring total protein signal directly on the membrane which was visualized before the chemiluminescent reaction and after the staining. In the Image Lab Software we downloaded both images and created a multichannel image of the stain-free and chemiluminescent blot images. Image Lab Software automatically detected lanes and bands. We selected a lane used for normalization (control sample, without treatment) and compared the chemiluminescent signal in each lane to its corresponding stain-free lane thus the normalized volume value represents each chemiluminescent signal that is properly total protein normalized to the stain-free signal in corresponding lane.
We found that this technology is reliable to accounts for variations during loading, electrophoresis, and transfer. More information concerning the technology can be found in the website of Bio-Rad. We also include an article comparing the traditional total protein normalization with stain-free technology.
We added appropriate explanation in the legend of figure 4 for the readers and in the section number 4.6. entitled CHOP protein expression analysis.
Gilda J and Gomes AV. (2013). Stain-Free total protein staining is a superior loading control to β-actin for Western blots. Anal. Biochem. Sept.15, 440 (2), 186–188.
Short R and Posch A. (2001). Stain-Free Approach for Western Blotting. Alternative to the Standard Blot Normalization Process. Genetic Engineering and Biotechnology News. Nov. 15, 31 (20). https://www.bio-rad.com/webroot/web/pdf/lsr/literature/Bulletin_RP0051.pdf
- All figure legends require the number of experiments done (n=?) and the statistical test
used.
Information about the statistical test used for assessing significance and the number of experiments was added.

Reviewer 3 Report
The work is devoted to the role of diclofenac on the tunicamycin-induced ER stress in endothelial cells. Specifically, the authors investigated the effect of diclofenac on gene expression in endothelial cells following ER stress. Diclofenac has been shown to inhibit stress-induced overexpression of the GRP78 gene and protein and has pro-survival effect of diclofenac on endothelial cells. However diclofenac had a slight effect on SHOP proteins expressions. The authors explained the contradictory data obtained in other works at other objects by using different concentrations of tunicamycin and diclofenac. The results obtained indicate the importance of stress intensity and dose dependence in treatment.
Remarks:
To further prove such important conclusions, it is necessary to repeat the experiment with a higher concentration of tunicamycin.
It is not clear why nothing is mentioned about the DNAJB9 proteins expressions
Author Response
Manuscript Number: molecules-1696198
Title: Diclofenac diminished the unfolded protein response (UPR) induced by tunicamycin in human endothelial cells.
At the very beginning we would like to thank the Reviewer for the great commitment and careful revision of our manuscript.
The work is devoted to the role of diclofenac on the tunicamycin-induced ER stress in endothelial cells. Specifically, the authors investigated the effect of diclofenac on gene expression in endothelial cells following ER stress. Diclofenac has been shown to inhibit stress-induced overexpression of the GRP78 gene and protein and has pro-survival effect of diclofenac on endothelial cells. However diclofenac had a slight effect on SHOP proteins expressions. The authors explained the contradictory data obtained in other works at other objects by using different concentrations of tunicamycin and diclofenac. The results obtained indicate the importance of stress intensity and dose dependence in treatment.
Remarks:
To further prove such important conclusions, it is necessary to repeat the experiment with a higher concentration of tunicamycin.
We agree with the Reviewer that a higher concentration of tunicamycin would give additional proof of the pro-survival effect of diclofenac. We would like to widen our research in the project which is being prepared and use different concentrations of tunicamycin and other ER stressors to check whether the action of NSAIDs depends on ER stress intensity. Different concentrations of tunicamycin and diclofenac shown in the literature data strongly depend on the type of cells used for experiments and chemical class of the drugs. Interestingly, in relation to the outcome of UPR activation some pharmacological discrepancy of NSAIDs can be observed between the results of experiments on the central nervous system (CNS) models showing activation of pro-survival UPR pathways [1-3] and experiments on cell cultures derived from the periphery which demonstrated mostly cytotoxic effects of NSAIDs due to induction of apoptotic UPR signalling [4-6]. However, such a generalization might be misleading and our short communication evaluating the role of diclofenac on the UPR signalling pathways activated by tunicamycin in endothelial cells seems to show some evidence. Therefore, we think it can be very interesting to prove in our next studies whether the effects of NSAIDs may depend on 1) ER stress intensity, 2) cell type and 3) chemical class of the drugs, even if some tendencies can be observed.
It is not clear why nothing is mentioned about the DNAJB9 proteins expressions
The DNAJB9 protein expressions were not studied. The goal of our study was to check the effect of diclofenac on ER stress-responsive genes responsible for activation of pro-survival or proapoptotic pathways. On the basis of the literature data we presumed that diclofenac may stimulate apoptosis. Unexpectedly, we observed that diclofenac significantly diminished activation of the UPR genes encoding both pro-survival (i.e. GRP78 and DNAJB9) and proapoptotic (i.e. CHOP) the UPR components. Therefore, we studied the effects of diclofenac on expressions of key UPR protein components, i.e. GRP78 and CHOP because in general there might be some differences between gene and protein expressions. We found that diclofenac significantly diminished GRP78 protein expression but had only minor and statistically insignificant influence on upregulation of CHOP protein which is a major regulator of apoptosis during ER stress. Thus, we also checked the impact of diclofenac on apoptosis induction and found that the drug significantly reduced the percentage of apoptotic endothelial cells induced by tunicamycin. These observations we would like to publish in a form of a short communication due to fact that the mechanism by which diclofenac may restore the UPR is not explained in detail.
References
- Llorente I.L., Burgin T.C., Perez-Rodriguez D., Martinez-Villayandre B., Perez-Garcia C.C., Fernandez-Lopez A., Unfolded protein response to global ischemia following 48 h of reperfusion in the rat brain: the effect of age and meloxicam, J Neurochem, 2013, 127(5), 701-10.
- Zhi Ye 1, Na Wang, Pingping Xia, E Wang, Juan Liao, Qulian Guo. Parecoxib suppresses CHOP and Foxo1 nuclear translocation, but increases GRP78 levels in a rat model of focal ischemia. Neurochem Res . 2013 Apr;38(4):686-93. doi: 10.1007/s11064-012-0953-4. Epub 2013 Jan 17.
- Yamazaki T., Muramoto M., Oe T., Morikawa N., Okitsu O., Nagashima T., et al., Diclofenac, a non-steroidal anti-inflammatory drug, suppresses apoptosis induced by endoplasmic reticulum stresses by inhibiting caspase signaling, Neuropharmacology, 2006, 50(5), 558-67.
- Tanaka K-I, Wataru Tomisato, Tatsuya Hoshino, Tomoaki Ishihara, Takushi Namba, Mayuko Aburaya, Takashi Katsu, Keitarou Suzuki, Shinji Tsutsumi, Tohru Mizushima. Involvement of intracellular Ca2+ levels in nonsteroidal anti-inflammatory drug-induced apoptosis. J Biol Chem . 2005 Sep 2;280(35):31059-67. doi: 10.1074/jbc.M502956200. Epub 2005 Jun 29.
- Nadanaciva, S,.;Aleo, M.D.; Strock, C.J.; Stedman, D.B.; Wang, H.; Will, Y. Toxicity assessments of nonsteroidal an-ti-inflammatory drugs in isolated mitochondria, rat hepatocytes, and zebrafish show good concordance across chemical classes. Toxicol Appl Pharmacol 2013, 272, 272-280.
- Franceschelli, S.; Moltedo, O.; Amodio, G.; Tajana, G.; Remondelli, P. In the Huh7 Hepatoma Cells Diclofenac and Indo-methacin Activate Differently the Unfolded Protein Response and Induce ER Stress Apoptosis. Open Biochem J 2011, 5, 45-51.
Reviewer 4 Report
Comments: Diclofenac diminished the unfolded protein response (UPR) induced by tunicamycin in human endothelial cells.
This manuscript, entitled "Diclofenac diminished the unfolded protein response (UPR) induced by tunicamycin in human endothelial cells" is observational, as the authors showed the experiments to validate the observation. However, the mechanistic validations have not been done. I hope these comments help in improvising the quality of the manuscript.
Major:
- The author is stating that diclofenac addition decreases the apoptotic cells. However, this must be evidenced by data measuring the mitochondrial functions and quality.
- Since ROS increases apoptosis, it is required to check the ROS in the presence of tunicamycin. Additionally, it would be required to check if the addition of diclofenac in the presence of tunicamycin reduces the ROS levels, thereby reducing the apoptosis, or diclofenac in the presence of tunicamycin decreases the apoptosis via some other mechanism.
- The author should take the other UPR inducers into account for comparing and confirmation of the data in the presence of diclofenac and know if it is a general UPR stressor response or specific to tunicamycin.
- The author only checked the expression of all selected genes (mentioned in Table 1) at the transcriptional level, except Grp78 and CHOP. It is suggested to check at the translational level because during ER stress, generally the phosphorylation of eIF2alpha gets increased, which leads to a reduction in the global translation. However, some selected genes are still being translated. It is advised that the authors be careful and not conclude the results on the basis of quantitative real time PCR.
- The authors lack the data to show how diclofenac restores the expression of Grp78 and CHOP and what would be the effect of diclofenac on the status of eIF2alpha phosphorylation.
Minor:
- All bar graphs should be replaced with graphs that explicitly show 3 or more data points with mean and SD values.
2. Page 2, line 53: The author should incorporate the reference for the statement.
3. Page 3, line 84: The author should correct the sentence.
Author Response
Manuscript Number: molecules-1696198
Title: Diclofenac diminished the unfolded protein response (UPR) induced by tunicamycin in human endothelial cells.
At the very beginning we would like to thank the Reviewer for the great commitment and accurate comments concerning our manuscript.
Comments: Diclofenac diminished the unfolded protein response (UPR) induced by tunicamycin in human endothelial cells.
This manuscript, entitled "Diclofenac diminished the unfolded protein response (UPR) induced by tunicamycin in human endothelial cells" is observational, as the authors showed the experiments to validate the observation. However, the mechanistic validations have not been done. I hope these comments help in improvising the quality of the manuscript.
We agree entirely with the Reviewer that results of our study are only observational. Therefore we would like to publish them as a short communication intended for reporting the preliminary results from a small investigation. On the basis of the literature data regarding the effects of NSAIDs on the activation of the UPR pathways we presumed that diclofenac may stimulate apoptosis. Unexpectedly, we observed that diclofenac significantly diminished activation of the UPR genes encoding both pro-survival and proapoptotic the UPR components. Additionally, the drug diminished significant upregulation and release of GRP78 protein but not CHOP protein and reduced apoptosis induction in endothelial cells. We think these results showed new observation therefore we would like to ask the Reviewer for considering publishing them in the present form.
Major:
- The author is stating that diclofenac addition decreases the apoptotic cells. However, this must be evidenced by data measuring the mitochondrial functions and quality.
- Since ROS increases apoptosis, it is required to check the ROS in the presence of tunicamycin. Additionally, it would be required to check if the addition of diclofenac in the presence of tunicamycin reduces the ROS levels, thereby reducing the apoptosis, or diclofenac in the presence of tunicamycin decreases the apoptosis via some other mechanism.
- The author should take the other UPR inducers into account for comparing and confirmation of the data in the presence of diclofenac and know if it is a general UPR stressor response or specific to tunicamycin.
- The author only checked the expression of all selected genes (mentioned in Table 1) at the transcriptional level, except Grp78 and CHOP. It is suggested to check at the translational level because during ER stress, generally the phosphorylation of eIF2alpha gets increased, which leads to a reduction in the global translation. However, some selected genes are still being translated. It is advised that the authors be careful and not conclude the results on the basis of quantitative real time PCR.
- The authors lack the data to show how diclofenac restores the expression of Grp78 and CHOP and what would be the effect of diclofenac on the status of eIF2alpha phosphorylation.
Minor:
- All bar graphs should be replaced with graphs that explicitly show 3 or more data points with mean and SD values.
Unfortunately, we do not understand how graphs should be prepared. We added the number of experiments (n) in descriptions of all figures.
- Page 2, line 53: The author should incorporate the reference for the statement.
The reference was added.
- Page 3, line 84: The author should correct the sentence.
The sentence was corrected.

Round 2
Reviewer 4 Report
Revised Report: Diclofenac diminished the unfolded protein response (UPR) induced by tunicamycin in human endothelial cells.
Comments have not been addressed and the authors have not done the necessary experiment to validate the observation needed in the manuscript as suggested in the previous revision.